# Dynamic-Depth Context Tree Weighting

**João V. Messias**[*]
Morpheus Labs
Oxford, UK
jmessias@morpheuslabs.co.uk

**Shimon Whiteson**
University of Oxford
Oxford, UK
shimon.whiteson@cs.ox.ac.uk

## Abstract

*Reinforcement learning* (RL) in partially observable settings is challenging because the agent's observations are not Markov. Recently proposed methods can learn variable-order Markov models of the underlying process but have steep memory requirements and are sensitive to aliasing between observation histories due to sensor noise. This paper proposes *dynamic-depth context tree weighting* (D2-CTW), a model-learning method that addresses these limitations. D2-CTW dynamically expands a suffix tree while ensuring that the size of the model, but not its depth, remains bounded. We show that D2-CTW approximately matches the performance of state-of-the-art alternatives at stochastic time-series prediction while using at least an order of magnitude less memory. We also apply D2-CTW to model-based RL, showing that, on tasks that require memory of past observations, D2-CTW can learn without prior knowledge of a good state representation, or even the length of history upon which such a representation should depend.

## 1 Introduction

Agents must often act given an incomplete or noisy view of their environments. While *decision-theoretic planning* and *reinforcement learning* (RL) methods can discover control policies for agents whose actions can have uncertain outcomes, *partial observability* greatly increases the problem difficulty since each observation does not provide sufficient information to disambiguate the true state of the environment and accurately gauge the utility of the agent's available actions. Moreover, when stochastic models of the system are not available *a priori*, probabilistic inference over latent state variables is not feasible. In such cases, agents must learn to memorize past observations and actions [21, 9], or one must learn history-dependent models of the system [15, 8].

*Variable-order Markov models* (VMMs), which have long excelled in stochastic time-series prediction and universal coding [23, 14, 2], have recently also found application in RL under partial observability [13, 7, 24, 19]. VMMs build a context-dependent predictive model of future observations and/or rewards, where a *context* is a variable-length subsequence of recent observations. Since the number of possible contexts grows exponentially with both the context length and the number of possible observations, VMMs' memory requirements may grow accordingly. Conversely, the frequency of each particular context in the data decreases as its length increases, so it may be difficult to accurately model long-term dependencies without requiring prohibitive amounts of data.

Existing VMMs address these problems by allowing models to differentiate between contexts at non-consecutive past timesteps, ignoring intermediate observations [13, 22, 10, 24, 4]. However, they typically assume that either the amount of input data is naturally limited or there is a known bound on the length of the contexts to be considered. In most settings in which an agent interacts continuously with its environment, neither assumption is well justified. The lack of a defined time limit means the approaches that make the former assumption, e.g., [13, 24], may eventually and indiscriminately

---

[*]During the development of this work, the main author was employed by the University of Amsterdam.

use all the agent's physical memory, while those that assume a bound on the context length, e.g., [19], may perform poorly if observations older than this bound are relevant.

This paper proposes *dynamic-depth context tree weighting* (D2-CTW), a VMM designed for general continual learning tasks. D2-CTW extends *context tree weighting* (CTW) [23] by allowing it to dynamically grow a suffix tree that discriminates between observations at different depths only insofar as that improves its ability to predict future inputs. This allows it to bound the number of contexts represented in the model, without sacrificing the ability to model long-term dependencies.

Our empirical results show that, when used for general stochastic time-series prediction, D2-CTW produces models that are much more compact than those of CTW while providing better results in the presence of noise. We also apply D2-CTW as part of a model-based RL architecture and show that it outperforms multiple baselines on the problem of RL under partial observability, particularly when an effective bound on the length of its contexts is not known *a priori*.

## 2 Background

### 2.1 Stochastic Time-Series Prediction

Let an *alphabet* $\Sigma = \{\sigma^1, \sigma^2, \ldots, \sigma^{|\Sigma|}\}$ be a discrete set of *symbols*, and let $\Pi(\Sigma)$ represent the space of probability distributions over $\Sigma$ (the $(|\Sigma|-1)$-simplex). Consider a discrete-time stochastic process that, at each time $t \geq 0$, samples a symbol $\sigma_t$ from a probability distribution $p_t \in \Pi(\Sigma)$. We assume that this stochastic process is stationary and ergodic, and that $p_t$ is a conditional probability distribution, which for some (unknown) constant integer $D$ with $0 < D \leq t$ has the form:

$$p_t(\sigma) = P(\sigma_t = \sigma \,|\, \sigma_{t-1}, \sigma_{t-2}, \ldots, \sigma_{t-D}). \tag{1}$$

Let $\boldsymbol{\sigma}_{t-D:t-1} = (\sigma_{t-D}, \sigma_{t-D+1}, \ldots \sigma_{t-1},)$ be a *string* of symbols from time $t - D$ to $t - 1$. Since $\boldsymbol{\sigma}_{t-D:t-1} \in \Sigma^D$ and $\Sigma$ is finite, there is a finite number of length-$D$ strings on which the evolution of our stochastic process can be conditioned. Thus, the stochastic process can also be represented by a time-invariant function $F : \Sigma^D \to \Pi(\Sigma)$ such that $p_t =: F(\boldsymbol{\sigma}_{t-D:t-1})$ at any time $t \geq D$.

Let $s$ be a string of symbols from alphabet $\Sigma$ with length $|s|$ and elements $[s]_{i=\in\{1,\ldots,|s|\}}$. Furthermore, a string $q$ with $|q| < |s|$ is said to be a *prefix* of $s$ iff $q_{1:|q|} = s_{1:|q|}$, and a *suffix* of $s$ iff $q_{1:|q|} = s_{|s|-|q|:|s|}$. We write $sq$ or $\sigma s$ for the concatenation of strings $s$ and $q$ or of $s$ and symbol $\sigma \in \Sigma$. A *complete and proper suffix set* is a set of strings $\mathcal{S}$ such that any string not in $\mathcal{S}$ has exactly one suffix in $\mathcal{S}$ but no string in $\mathcal{S}$ has a suffix in $\mathcal{S}$.

Although $D$ is an upper bound on the age of the oldest symbol on which the process $F$ depends, at any time $t$ it may depend only on some suffix of $\boldsymbol{\sigma}_{t-D:t-1}$ of length less than $D$. Given the variable-length nature of its conditional arguments, $F$ can be tractably encoded as a $D$-bounded *tree source* [2] that arranges a complete and proper suffix set into a tree-like graphical structure. Each node at depth $d \leq D$ corresponds to a length-$d$ string and all internal nodes correspond to suffixes of the strings associated with their children; and each leaf encodes a distribution over $\Sigma$ representing the value of $F$ for that string.

Given a single, uninterrupted sequence of $\boldsymbol{\sigma}_{0:t}$ generated by $F$, we wish to learn the $\tilde{F} : \Sigma^D \to \Pi(\Sigma)$ that minimises the *average log-loss* of the observed data $\boldsymbol{\sigma}_{0:t}$. Letting $P_{\tilde{F}}(\cdot \,|\, \sigma_{i-D}, \ldots, \sigma_{i-1}) := \tilde{F}(\boldsymbol{\sigma}_{i-D:i-1})$:

$$l(\boldsymbol{\sigma}_{0:t} \,|\, \tilde{F}) = -\frac{1}{t} \sum_{i=D}^{t} \log P_{\tilde{F}}(\sigma_i \,|\, \sigma_{i-D}, \ldots, \sigma_{i-1}). \tag{2}$$

### 2.2 Context Tree Weighting

The depth-$K$ *context tree* on alphabet $\Sigma$ is a graphical structure obtained by arranging all possible strings in $\Sigma^K$ into a full tree. A context tree has a fixed depth at all leaves and potentially encodes *all* strings in $\Sigma^K$, not just those required by $F$. More specifically, given a sequence of symbols $\boldsymbol{\sigma}_{0:t-1}$, the respective length-$K$ *context* $\boldsymbol{\sigma}_{t-K:t-1}$ induces a *context path* along the context tree by following at each level $d \leq K$ the edge corresponding to $\sigma_{t-d}$. The *root* of the context tree represents an empty string $\emptyset$, a suffix to all strings. Furthermore, each node keeps track of the input symbols that have immediately followed its respective context. Let $sub(\boldsymbol{\sigma}_{0:t-1}, s)$ represent the

string obtained by concatenating all symbols $\sigma_i$ in $\boldsymbol{\sigma}_{0:t-1}$ such that its preceding symbols verify $\sigma_{i-k} = s_k$ for $k = 1, \dots, |s|$. Then, each node $s$ in the context tree maintains its own estimate of the probability of observing the string $sub(\boldsymbol{\sigma}_{0:t-1}, s)$.

*Context tree weighting* (CTW) [23] learns a mixture of the estimates of $P(sub(\boldsymbol{\sigma}_{0:t-1}, s))$ at all contexts $s$ of length $|s| \leq K$ and uses it to estimate the probability of the entire observed sequence. Let $P_e^s(\boldsymbol{\sigma}_{0:t-1})$ represent the estimate of $P(sub(\boldsymbol{\sigma}_{0:t-1}, s))$ at the node corresponding to $s$, and let $P_w^s(\boldsymbol{\sigma}_{0:t-1})$ be a *weighted* representation of the same measure, defined recursively as:

$$P_w^s(\boldsymbol{\sigma}_{0:t-1}) := \begin{cases} \frac{1}{2} P_e^s(\boldsymbol{\sigma}_{0:t-1}) + \frac{1}{2} \prod_{\sigma \in \Sigma} P_w^{\sigma s}(\boldsymbol{\sigma}_{0:t-1}) & \text{if } |s| < K, \\ P_e^s(\boldsymbol{\sigma}_{0:t-1}) & \text{if } |s| = K. \end{cases} \quad (3)$$

Since $sub(\boldsymbol{\sigma}_{0:t-1}, \emptyset) = \boldsymbol{\sigma}_{0:t-1}$ by definition of the empty context, $P_w^\emptyset(\boldsymbol{\sigma}_{0:t-1})$ is an estimate of $P(\boldsymbol{\sigma}_{0:t-1})$. The conditional probability of symbol $\sigma_t$ is approximated as $P_{\tilde{F}}(\sigma_t|\boldsymbol{\sigma}_{0:t-1}) = P_w^\emptyset(\boldsymbol{\sigma}_{0:t})/P_w^\emptyset(\boldsymbol{\sigma}_{0:t-1})$.

The (unweighted) estimate $P_e^s(\boldsymbol{\sigma}_{0:t-1})$ at each context is often computed by keeping $|\Sigma|$ incrementally updated counters $[c_{s,t}]_{i=1,\dots,|\Sigma|} \in \mathbb{N}_0$, where for each $\sigma^i \in \Sigma$, $[c_{s,t}]_i$ represents the total number of instances where the substring $s\sigma^i$ can be found within $\boldsymbol{\sigma}_{0:t-1}$. The vector of counters $\mathbf{c}_{s,t}$ can be modelled as the output of a Dirichlet-multinomial distribution with *concentration parameter* vector $\boldsymbol{\alpha} = [\alpha_i]_{i=1,\dots,|\Sigma|}$. An estimate of the probability of observing symbol $\sigma^k$ at time $t+1$ can then be taken as follows: if $s$ is on the context path at time $t$ and $\sigma_t = \sigma^k$ is the next observed symbol, then $[c_{s,t+1}]_k = [c_{s,t}]_k + 1$, and $[c_{s,t+1}]_i = [c_{s,t}]_i$ for all $i \neq k$. Then:

$$P_e^s(\sigma^k | \boldsymbol{\sigma}_{0:t}) := \frac{P_{DirM}(\mathbf{c}_{s,t+1} | \boldsymbol{\alpha})}{P_{DirM}(\mathbf{c}_{s,t} | \boldsymbol{\alpha})} = \frac{[c_{s,t}]_k + [\alpha]_k}{\mathbf{c}_{s,t}^+ + \boldsymbol{\alpha}^+}, \quad (4)$$

where $\boldsymbol{\alpha}^+ = \sum_{i=1}^{|\Sigma|} [\alpha]_i$, $\mathbf{c}_{s,t}^+ = \sum_{i=1}^{|\Sigma|} [c_{s,t}]_i$, and $P_{DirM}$ is the Dirichlet-multimomial mass function. The estimate of the probability of the full sequence is then $P_e^s(\boldsymbol{\sigma}_{0:t}) = \prod_{\tau=0}^t P_e^s(\sigma_\tau|\boldsymbol{\sigma}_{0:\tau-1})$. This can be updated in constant time as each new symbol is received. The choice of $\boldsymbol{\alpha}$ affects the overall quality of the estimator. We use the *sparse adaptive Dirichlet* (SAD) estimator [11], which is especially suited to large alphabets.

In principle, a depth-$K$ context tree has $|\Sigma|^{K+1} - 1$ nodes, each with at most $|\Sigma|$ integer counters. In practice, there may be fewer nodes since one need only to allocate space for contexts found in the data at least once, but their total number may still grow linearly with the length of the input string. Thus, for problems such as partially observable RL, in which the amount of input data is unbounded, or for large $|\Sigma|$ and $K$, the memory used by CTW can quickly become unreasonable.

Previous extensions to CTW and other VMM algorithms have been made that do not explicitly bound the depth of the model [6, 22]. However, these still take up memory that is worst-case linear in the length of the input sequence. Therefore, they are not applicable to reinforcement learning. To overcome this problem, most existing approaches artificially limit $K$ to a low value, which limits the agent's ability to address long-term dependencies. To our knowledge, the only existing principled approach to reducing the amount of memory required by CTW was proposed in [5], through the use of a modified (*Budget*) SAD estimator which can be used to limit the branching factor in the context tree to $B < |\Sigma|$. This approach still requires $K$ to be set *a priori*, and is best-suited to prediction problems with large alphabets but few high frequency symbols (*e.g.* word prediction), which is not generally the case in decision-making problems.

## 2.3 Model-Based RL with VMMs

In RL with partial observability, an agent performs at each time $t$ an *action* $a_t \in \mathcal{A}$, and receives an *observation* $o_t \in \mathcal{O}$ and a *reward* $r_t \in \mathbb{R}$ with probabilities $P(o_t|o_{0:t-1}, r_{0:t-1}, a_{0:t-1})$ and $P(r_t|o_{0:t-1}, r_{0:t-1}, a_{0:t-1})$ respectively. This representation results from marginalising out the latent state variables and assuming that the agent observes rewards. The agent's goal is to maximise the expected cumulative future rewards $E\{\sum_{\tau=t+1}^\infty r_t + \lambda^{\tau-t} r_\tau\}$ for some *discount factor* $\lambda \in [0, 1)$.

Letting $\mathcal{R} = \{r_t : P(r_t|o_{0:t-1}, r_{0:t-1}, a_{0:t-1}) > 0 \,\forall o_{0:t-1}, r_{0:t-1}, a_{0:t-1}\}$ represent the set of possible rewards and $z_t \in \{1, \dots, |\mathcal{R}|\}$ the unique index of $r_t \in \mathcal{R}$, then a *percept* $(o_t, z_t)$ is received at each time with probability $P(o_t, z_t|o_{0:t-1}, z_{0:t-1}, a_{0:t-1})$. VMMs such as CTW can

then learn a model of this process, using the alphabet $\Sigma = \mathcal{O} \times \{1, \ldots, |\mathcal{R}|\}$. This predictive model must condition on past actions, but its output should only estimate the probability of the next percept (not the next action). This is solved by interleaving actions and percepts in the input context, but only updating its estimators based on the value of the next percept [19]. The resulting *action-conditional* model can be used as a simulator by sample-based planning methods such as UCT [12].

## 2.4 Utile Suffix Memory

*Utile suffix memory* (USM) [13] is an RL algorithm similar to VMMs for stochastic time-series prediction. USM learns a suffix tree that is conceptually similar to a context tree with the following differences. First, each node in the suffix tree directly maintains an estimate of expected cumulative future reward for each action. To compute this estimate, USM still predicts (immediate) future observations and rewards at each context, analogously to VMM methods. This prediction is done in a purely frequentist manner, which often yields inferior prediction performance compared to other VMMs, especially given noisy data.

Second, USM's suffix tree does not have a fixed depth; instead, its tree is grown incrementally, by testing potential expansions for statistically significant differences between their respective predictions of cumulative future reward. USM maintains a fixed-depth subtree of *fringe nodes* below the proper leaf nodes of the suffix tree. Fringe nodes do not contribute to the model's output, but they also maintain count vectors. At regular intervals, USM compares the distributions over cumulative future reward of each fringe node against its leaf ancestor, through a *Kolmogorov-Smirnov* (K-S) test. If this test succeeds at some threshold confidence, then *all* fringe nodes below that respective leaf node become proper nodes, and a new fringe subtree is created below the new leaf nodes.

USM's fringe expansion allows it to use memory efficiently, as only the contextual distinctions that are actually significant for prediction are represented. However, USM is computationally expensive. Performing K-S tests for all nodes in a fringe subtree requires, in the worst-case, time linear in the amount of (real-valued) data contained at each node, and exponential in the depth of the subtree. This cost can be prohibitive even if the expansion test is only run infrequently. Furthermore, USM does not explicitly bound its memory use, and simply stopping growth once a memory bound is hit would bias the model towards symbols received early in learning.

## 3 Dynamic-Depth Context Tree Weighting

We now propose *dynamic-depth context tree weighting* (D2-CTW). Rather than fixing the depth *a priori*, like CTW, or using unbounded memory, like USM, D2-CTW learns $\tilde{F}$ with dynamic depth, subject to the constraint $|\tilde{F}_t| \leq L$ at any time $t$, where $L$ is a fixed *memory bound*.

### 3.1 Dynamic Expansion in CTW

To use memory efficiently and avoid requiring a fixed depth, we could simply replicate USM's fringe expansion in CTW, by performing K-S tests on distributions over symbols ($P_e^s$) instead of distributions over expected reward. However, doing so would introduce bias. The weighted estimates $P_w^s(\boldsymbol{\sigma}_{0:t})$ for each context $s$ depend on the ratio of the probability of the observed data at $s$ itself, $P_e^s(\boldsymbol{\sigma}_{0:t})$, and that of the data observed at its children, $P_w^{s'}(\boldsymbol{\sigma}_{0:t})$ at $s' = \sigma s \, \forall \sigma \in \Sigma$. These estimates depend on the number of times each symbol followed a context, implying that $\mathbf{c}_{s,t} = \sum_{\sigma \in \Sigma} \mathbf{c}_{s',t}$. Thus, the weighting in (3) assumes that each symbol that was observed to follow the non-leaf context $s$ was also observed to follow exactly one of its children $s'$. If this was not so and, e.g., $s$ was created at time 0 but its children only at $\tau > 0$, then, since $P_w^{s'}(\boldsymbol{\sigma}_{\tau:t}) \geq P_w^{s'}(\boldsymbol{\sigma}_{0:t})$, the weighting would be biased towards the children, which would have been exposed to less data.

Fortunately, an alternative CTW recursion, originally proposed for numerical stability [20], overcomes this issue. In CTW and for a context tree of fixed depth $K$, let $\beta_t^s$ be the likelihood ratio between the weighted estimate below $s$ and the local estimate at $s$ itself:

$$\beta_t^s := \begin{cases} \frac{\prod_{\sigma \in \Sigma} P_w^{\sigma s}(\boldsymbol{\sigma}_{0:t})}{P_e^s(\boldsymbol{\sigma}_{0:t})} & \text{if } |s| < K, \\ 1 & \text{if } |s| = K. \end{cases} \tag{5}$$

Then, the weighted estimate of the *conditional* probability of an observed symbol $\sigma_t$ at node $s$ is:

$$P_w^s(\sigma_t|\boldsymbol{\sigma}_{0:t-1}) := \frac{P_w^s(\boldsymbol{\sigma}_{0:t})}{P_w^s(\boldsymbol{\sigma}_{0:t-1})} = \frac{\frac{1}{2}P_e^s(\boldsymbol{\sigma}_{0:t})\left(1 + \beta_t^s\right)}{\frac{1}{2}P_e^s(\boldsymbol{\sigma}_{0:t-1})\left(1 + \beta_{t-1}^s\right)} =: P_e^s(\sigma_t|\boldsymbol{\sigma}_{0:t-1})\frac{1 + \beta_t^s}{1 + \beta_{t-1}^s}. \tag{6}$$

Furthermore, $\beta_t^s$ can be updated for each $s$ as follows. Let $\mathcal{C}_t$ represent the set of suffixes on the context path at time $t$ (the set of all suffixes of $\boldsymbol{\sigma}_{0:t-1}$). Then:

$$\beta_t^s = \begin{cases} \frac{P_w^{s'}(\sigma_t|\boldsymbol{\sigma}_{0:t-1})}{P_e^s(\sigma_t|\boldsymbol{\sigma}_{0:t-1})}\beta_{t-1}^s & \text{if } s \in \mathcal{C}_t, \\ \beta_t^s = \beta_{t-1}^s & \text{otherwise,} \end{cases} \tag{7}$$

where $s' = \sigma_{t-1-|s|}s$ is the child of $s$ that follows it on the context path. For any context, we set $\beta_0^s = 1$. This reformulation allows the computation of $P_w^s(\sigma_t|\boldsymbol{\sigma}_{0:t-1})$ using only the nodes on the context path and while storing only a single value in those nodes, $\beta_t^s$, regardless of $|\Sigma|$.

Since this reformulation depends only on conditional probability estimates, we can perform fringe expansion in CTW and add nodes dynamically without biasing the mixture. Disregard the fixed depth limit $K$ and consider instead a suffix tree where all leaf nodes have a depth greater than the *fringe depth* $H > 0$. For any leaf node at depth $d$, its ancestor at depth $d - H$ is its *frontier node*. The descendants of any frontier node are *fringe nodes*. Let $f_t$ represent the frontier node on the context path at time $t$. At every timestep $t$, we traverse down the tree by following the context path as in CTW. At every node on the context path and above $f_t$, we apply (6) and (7) while treating $f_t$ as a leaf node. For $f_t$ and the fringe nodes on the context path below it, we apply the same updates while treating fringe nodes normally. Thus, the recursion in (6) does not carry over to fringe nodes, but otherwise all nodes update their values of $\beta$ in the same manner.

Once the fringe expansion criterion is met (see Section 3.2), the fringe nodes below $f_t$ simply stop being labeled as such, while the values of $\beta$ for the nodes above $f_t$ must be updated to reflect the change in the model. Let $\bar{P}_w^{f_t}(\boldsymbol{\sigma}_{0:t})$ represent the weighted (unconditional) output at $f_t$ *after* the fringe expansion step. We have therefore $\bar{P}_w^{f_t}(\boldsymbol{\sigma}_{0:t}) := \frac{1}{2}P_e^{f_t}(\boldsymbol{\sigma}_{0:t})(1 + \beta_t^{f_t})$, but *prior* to the expansion, $P_w^{f_t}(\boldsymbol{\sigma}_{0:t}) = P_e^{f_t}(\boldsymbol{\sigma}_{0:t})$. The net change in the likelihood of $\boldsymbol{\sigma}_{0:t}$, according to $f_t$, is:

$$\alpha_{exp}^{f_t} := \frac{\bar{P}_w^{f_t}(\boldsymbol{\sigma}_{0:t})}{P_w^{f_t}(\boldsymbol{\sigma}_{0:t})} = \frac{1 + \beta_t^{f_t}}{2}. \tag{8}$$

This induces a change in the likelihood of the data according to all of the ancestors of $f_t$. We need to determine $\alpha_{exp}^{\emptyset} =: \bar{P}_w^{\emptyset}(\boldsymbol{\sigma}_{0:t})/P_w^{\emptyset}(\boldsymbol{\sigma}_{0:t})$, which quantifies the effect of the fringe expansion on the global output of the weighted model.

**Proposition 1.** *Let $f$ be a string corresponding to a frontier node, and let $p_d$ be the length-$d$ suffix of $f$ (with $p_0 = \emptyset$). Also let $\rho^f := \prod_{d=0}^{|f|-1} \frac{\beta_t^{p_d}}{1+\beta_t^{p_d}}$, and $\alpha_{exp}^f := \frac{1+\beta_t^f}{2}$. Then:*

$$\alpha_{exp}^{\emptyset} := \frac{\bar{P}_w^{\emptyset}(\boldsymbol{\sigma}_{0:t})}{P_w^{\emptyset}(\boldsymbol{\sigma}_{0:t})} = 1 + \rho^f \left(\alpha_{exp}^f - 1\right).$$

The proof can be found in the supplementary material of this paper (Appendix A.1). This formulation is useful since, for any node $s$ in the suffix tree with ancestors $(p_0, p_1, \ldots, p_{|s|-1})$ we can associate a value $\rho_t^s = \prod_{d=0}^{|s|-1} \frac{\beta_t^{p_d}}{1+\beta_t^{p_d}} = \rho_t^{p_{|s|-1}} \frac{\beta_t^{p_{|s|-1}}}{1+\beta_t^{p_{|s|-1}}}$ that measures the sensitivity of the whole model to changes below $s$, and not necessarily just fringe expansions. Thus, a node with $\rho^s \simeq 0$ is a good candidate for pruning (see Section 3.3). Furthermore, this value can be computed while traversing the tree along the context path. Although the computation of $\rho^s$ for a particular node still requires $O(|s|)$ operations, the values of $\rho$ for all ancestors of $s$ are also computed along the way.

## 3.2 Fringe Expansion Criterion

As a likelihood ratio, $\alpha_{exp}^f$ provides a statistical measure of the difference between the predictive model at each frontier node $f$ and that formed by its fringe children. Analogously, $\alpha_{exp}^{\emptyset}$ can be seen as the likelihood ratio between two models that differ only on the subtree below $f$. Therefore, we can test the hypothesis that the subtree below $f$ should be added to the model by checking if $\alpha_{exp}^{\emptyset} > \gamma$ for some $\gamma > 1$. Since the form of $P_w^{\emptyset}(\cdot)$ is unknown, we cannot establish proper confidence levels for $\gamma$; however, the following result shows that the value of $\gamma$ is not especially important, since if the subtree below $f$ improves the model, this test will eventually be true given enough data.

**Theorem 1.** *Let $\mathcal{S}$ and $\mathcal{S}_{exp}$ be two proper suffix sets such that $\mathcal{S}_{exp} = (\mathcal{S} \setminus f) \cup \mathcal{F}$ where $f$ is suffix to all $f' \in \mathcal{F}$. Furthermore, let $M$ and $M_{exp}$ be the CTW models using the suffix trees induced by $\mathcal{S}$ and $\mathcal{S}_{exp}$ respectively, and $P_w^{\emptyset}(\boldsymbol{\sigma}_{0:t}; M)$, $P_w^{\emptyset}(\boldsymbol{\sigma}_{0:t}; M_{exp})$ their estimates of the likelihood of $\boldsymbol{\sigma}_{0:t}$.*

*If there is a $T \in \mathbb{N}$ such that, for any $\tau > 0$:*

$$\prod_{t=\tau}^{T+\tau} P_e^f(\sigma_t | \boldsymbol{\sigma}_{0:t-1}; M) < \prod_{t=\tau}^{T+\tau} \prod_{\sigma \in \Sigma} P_w^{\sigma f}(\sigma_t | \boldsymbol{\sigma}_{0:t-1}; M_{exp}),$$

*then for any $\gamma \in [1, \infty)$, there is $T' > 0$ such that $P_w^{\emptyset}(\boldsymbol{\sigma}_{0:T'}; M_{exp})/P_w^{\emptyset}(\boldsymbol{\sigma}_{0:T'}; M) > \gamma$.*

The proof can be found in the supplementary material (Appendix A.2). Using $\alpha_{exp}^{\emptyset} > \gamma$ as a statistical test instead of K-S tests yields great computational savings, since the procedure described in Proposition 1 allows us to determine this test in $O(|f_t|)$, typically much lower than the $O(|\Sigma|^{H+1})$ complexity of K-S testing all fringe children.

Theorem 1 also ensures that, if sufficient memory is available, D2-CTW will eventually perform as well as CTW with optimal depth bound $K = D$. This follows from the fact that, for every node $s$ at depth $d_s \leq D$ in a CTW suffix tree, if $\beta_t^s \geq 1$ for all $t > \tau$, then the D2-CTW suffix tree will be at least as deep as $d_s$ at context $s$ after some time $t' \geq \tau$. That is, at some point, the D2-CTW model will contain the "useful" sub-tree of the optimal-depth context tree.

**Corollary 1.** *Let $l(\cdot | \tilde{F}_{CTW}, D)$ represent the average log-loss of CTW using fixed depth $K$ when modeling a $D-$bounded tree source, and $l(\cdot | \tilde{F}_{D2-CTW}, \gamma, H, L)$ the same metric when using D2-CTW. For any values of $\gamma > 1$ and $H > 1$, and for sufficiently high $L > 0$, there exists a time $T' > 0$ such that, for any $t > T'$, $l(\boldsymbol{\sigma}_{T':t} | \tilde{F}_{D2-CTW}, \gamma, H, L) \leq l(\boldsymbol{\sigma}_{T':t} | \tilde{F}_{CTW}, D)$.*

### 3.3 Ensuring the Memory Bound

In order to ensure that the memory bound $|\tilde{F}_t| \leq L$ is respected, we must first consider whether a potential fringe expansion does not require more memory than is available. Thus, if the subtree below frontier node $f$ has size $L_f$, we must test if $|\tilde{F}_t| + L_f \leq L$. This means that fringe nodes are not taken into account when computing $|\tilde{F}_t|$, as they do not contribute to the output of $\tilde{F}_t$ and are therefore considered as memory overhead, and discarded after training.

Once $|\tilde{F}_t|$ is such that no fringe expansions are possible without violating the memory bound, it may still be possible to improve the model by pruning low-quality subtrees to create enough space for more valuable fringe expansions. Pruning operations also have a quantifiable effect on the likelihood of the observed data according to $\tilde{F}_t$. Let $\underline{P}_w^s(\boldsymbol{\sigma}_{0:t})$ represent the weighted estimate at internal node $s$ after pruning its subtree. Analogously to (8), we can define $\alpha_{prune}^s := \underline{P}_w^s(\boldsymbol{\sigma}_{0:t})/P_w^s(\boldsymbol{\sigma}_{0:t}) = 2/(1 + \beta_t^s)$. We can also compute $\alpha_{prune}^{\emptyset}$, the global effect on the likelihood, using the procedure in Proposition 1. Since $\alpha_{prune}^{\emptyset} = 1 + \rho^s (\alpha_{prune}^s - 1)$, typically with $\alpha_{prune}^s < 1$, if a fringe expansion at $f$ increases $P_w^{\emptyset}(\boldsymbol{\sigma}_{0:t})$ by a factor of $\alpha_{exp}^{\emptyset}$ but requires space $L_f$ such that $|\tilde{F}_t| + L_f > L$, we should prune the subtree below $s \neq f$ that frees $L_s$ space and reduces $P_w^{\emptyset}(\boldsymbol{\sigma}_{0:t})$ by $\alpha_{prune}^{\emptyset}$ if 1) $\alpha_{exp}^{\emptyset} \times \alpha_{prune}^{\emptyset} > 1$; 2) $|\tilde{F}_t| + L_f - L_s \leq L$; and 3) $s$ is not an ancestor of $f$. The latter condition requires $O(|f| - |s|)$ time to validate, while the former can be done in constant time if $\rho^s$ is available.

In general, some combination of subtrees could be pruned to free enough space for some combination of fringe expansions, but determining the best possible combination of operations at each time is too computationally expensive. As a tractable approximation, we compare only the best single expansion and prune at nodes $f^*$ and $s^*$ respectively, quantified with two heuristics $\mathcal{H}_{exp}^f := \log \alpha_{expf}^{\emptyset}$ and $\mathcal{H}_{prune}^s := -\log \alpha_{prune_s}^{\emptyset}$, such that $f^* = \arg \max_f \mathcal{H}_{exp}^f$ and $s^* = \arg \min_f \mathcal{H}_{prune}^s$.

As $L$ is decreased, the performance of D2-CTW may naturally degrade. Although Corollary 1 may no longer be applicable in that case, a weaker bound on the performance of memory-constrained D2-CTW can be obtained as follows, regardless of $L$: let $d_{min}^t$ denote the minimum depth of any frontier node at time $t$; then the D2-CTW suffix tree covers the set of $d_{min}^t$-bounded *models* [23]. The *redundancy* of D2-CTW, measured as the Kullback-Leibler divergence $D_{KL}(F || \tilde{F}_t)$, is then at least as low as the redundancy of a multi-alphabet CTW implementation with $K = d_{min}^t$ [17].

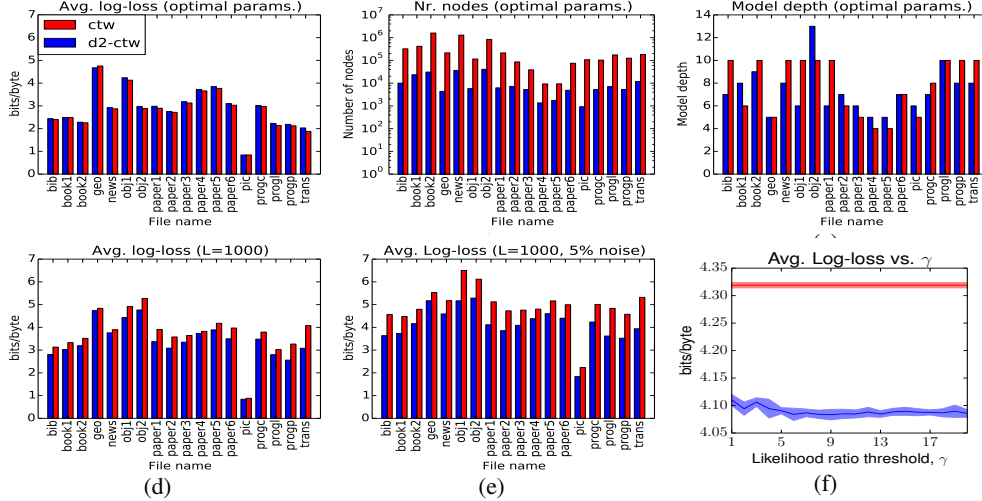

Figure 1: *Calgary Corpus* performance with CTW (red) and D2-CTW (blue). For average log-loss, lower is better: (a)-(c) using optimal parameters; (d) with a bound on the number of nodes; (e) with size bound and uniform noise; (f) log-loss vs. $\gamma$ on 'book2', with $10\%$ noise (over 30 runs).

### 3.4 Complete Algorithm and Complexity

The complete D2-CTW algorithm operates as follows (please refer to Appendix A.3 for the respective pseudo-code): a suffix tree is first initialized containing only a root node; at every timestep, the suffix tree is updated using the observed symbol $\sigma_t$, and the preceding context (if it exists) from time $t - d_{max}^t - H$ where $d_{max}^t$ is the current maximum depth of the tree and $H$ is the fringe depth. This update returns the weighted conditional probability of $\sigma_t$, and it also keeps track of the best known fringe expansion and pruning operations. Then, a post-processing step expands and possibly prunes the tree as necessary, ensuring the memory bound is respected. This step also corrects the values of $\beta$ for any nodes affected by these topological operations. D2-CTW trains on each new symbol in $O(d_{max}^t + H)$ time, the same as CTW with depth bound $K = d_{max}^t + H$. A worst-case $O((d_{max}^t + H)|\Sigma|)$ operations are necessary to sample a symbol from the learned model, also equivalent to CTW. Post-processing requires $O(\max\{|f^*|, |s^*|\})$ time.

## 4 Experiments

We now present empirical results on byte-prediction tasks and partially-observable RL. Our code and instructions for its use is publicly available at: `https://bitbucket.org/jmessias/vmm_py`.

**Byte Prediction** We compare the performance of D2-CTW against CTW on the 18-file variant of the *Calgary Corpus* [3], a benchmark of text and binary data files. For each file, we ask the algorithms to predict the next byte given the preceding data, such that $|\Sigma| = 256$ across all files.

We first compare performance when using (approximately) optimal hyperparameters. For CTW, we performed a grid search taking $K \in \{1, \dots, 10\}$ for each file. For D2-CTW, we investigated the effect of $\gamma$ on the prediction log-loss across different files, and found no significant effect of this parameter for sufficiently large values (an example is shown in Fig. 1f), in accordance with Theorem 1. Consequently, we set $\gamma = 10$ for all our D2-CTW runs. We also set $L = \infty$ and $H = 2$.

The corpus results, shown in Figs. 1a–1c, show that D2-CTW achieves comparable performance to CTW: on average D2-CTW's loss is $2\%$ higher, which is expected since D2-CTW grows dynamically from a single node, while CTW starts with a fully grown model of optimal height. By contrast, D2-CTW uses many fewer nodes than CTW, by at least one order of magnitude (average factor $\sim 28$). D2-CTW automatically discovers optimal depths that are similar to the optimal values for CTW. We then ran a similar test but with a bound on the number of nodes $L = 1000$. For CTW, we enforced this bound by simply stopping the suffix tree from growing beyond this point[2]. The results

are shown in Fig. 1d. In this case, the log-loss of CTW is on average $11.4\%$ and up to $32.3\%$ higher than that of D2-CTW, showing that D2-CTW makes a significantly better use of memory.

Finally, we repeated this test but randomly replaced $5\%$ of symbols with uniform noise. This makes the advantage of D2-CTW is even more evident, with CTW scoring on average $20.0\%$ worse (Fig. 1e). While the presence of noise still impacts performance, the results show that D2-CTW, unlike CTW, is resilient to noise: spurious contexts are not deemed significant, avoiding memory waste.

**Model-Based RL**  For our empirical study on online partially observable RL tasks, we take as a baseline MC-AIXI, a combination of fixed-depth CTW modelling with $\rho$UCT planning [19], and investigate the effect of replacing CTW with D2-CTW and limiting the available memory. We also compare against PPM-C, a frequentist VMM that is competitive with CTW [2]. Our experimental domains are further described in the supplementary material.

Our first domain is the *T-maze* [1], in which an agent must remember its initial observation in order to act optimally at the end of the maze. We consider a maze of length $4$. We set $K = 3$ for CTW and PPM-C, which is the guaranteed minimum depth to produce the optimal policy. For D2-CTW we set $\gamma = 1$, $H = 2$, and do not enforce a memory bound. As in [19], we use an $\epsilon$-greedy exploration strategy. Fig. 2a shows that D2-CTW discovers the length of the T-Maze automatically. Furthermore, CTW and PPM-C fail to learn to retain the required observations, as during the initial stages of learning the agent may need more than 3 steps to reach the goal (D2-CTW learns a model of depth $4$).

Our second scenario is the *cheese maze* [13], a navigation task with aliased observations. Under optimal parameters, D2-CTW and CTW both achieve near-optimal performance for this task. We investigated the effect of setting a bound on the number of nodes $L = 1000$, roughly $1/5$ of the amount used by CTW with optimal hyperparameters. In Fig. 2b we show that the quality of D2-CTW degrades less than both CTW and PPM-C, still achieving a near optimal policy. As this is a small-sized problem with $D = 2$, CTW and PPM-C still produce reasonable results in this case albeit with lower quality models than D2-CTW.

Finally, we tested a partially observable version of *mountain car* [16], in which the position of the car is observed but not its velocity. We coarsely discretised the position of the car into $10$ states. In this task, we have no strong prior knowledge about the required context length, but found $K = 4$ to be sufficient for optimal PPM-C and CTW performance. For D2-CTW, we used $\gamma = 10$, $H = 2$. We set also $L = 1000$ for all methods. Fig. 2c shows the markedly superior performance of D2-CTW when subject to this memory constraint.

## 5 Conclusions and Future Work

We introduced D2-CTW, a variable-order modelling algorithm that extends CTW by using a fringe expansion mechanism that tests contexts for statistical significance, and by allowing the dynamic adaptation of its suffix tree while subject to a memory bound. We showed both theoretically and empirically that D2-CTW requires little configuration across domains and provides better performance with respect to CTW under memory constraints and/or the presence of noise. In future work, we will investigate the use the Budget SAD estimator with a dynamic budget as an alternative mechanism for informed pruning. We also aim to apply a similar approach to *context tree switching* (CTS) [18], an algorithm that is closely related to CTW but enables mixtures in a larger model class.

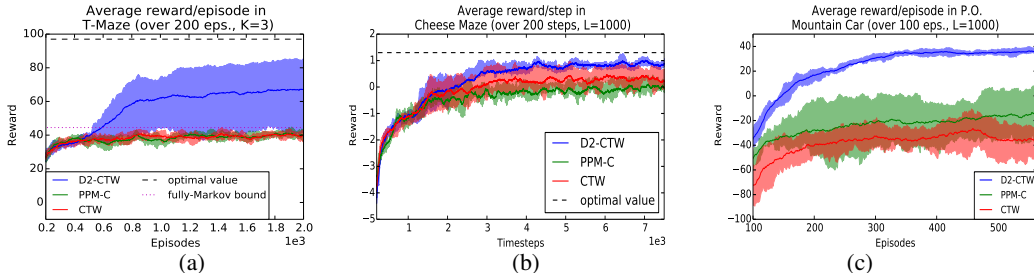

Figure 2: Performance measured as (running) average rewards in (a) *T-maze*; (b) *cheese maze*; (c) partially observable *mountain car*. Results show mean over 10 runs, and shaded first to third quartile.

**Acknowledgments**

This work was supported by the European Commission under the grant agreement FP7-ICT-611153 (TERESA).

## Footnotes

[2] For simplicity, we did not use CTW with Budget SAD as a baseline. Budget SAD could also be used to extend D2-CTW, so a fair comparison would necessitate the optional integration of Budget SAD into both CTW and D2-CTW. This is an interesting possibility for future work.

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
