[Supplementary Material]

# Dynamic-Depth Context Tree Weighting: Supplementary Material

## A   Appendices

### A.1   Proof of Theorem 1

Let $k \in \mathbb{N}$ and $T' = kT$. Then,

$$P_e^f(\boldsymbol{\sigma}_{0:T'}) = \prod_{i=0}^{k-1} P_e^f(\boldsymbol{\sigma}_{iT:(i+1)T} | \boldsymbol{\sigma}_{0:iT-1})$$

$$= \prod_{i=1}^{k} \prod_{t=iT}^{T+iT} P_e^f(\sigma_t | \boldsymbol{\sigma}_{0:t-1}) \Leftrightarrow$$

$$\Leftrightarrow \frac{\prod\limits_{\sigma \in \Sigma} P_w^{\sigma f}(\boldsymbol{\sigma}_{0:T'})}{P_e^f(\boldsymbol{\sigma}_{0:T'})} = \prod_{i=1}^{k} \prod_{t=iT}^{T+iT} \frac{\prod\limits_{\sigma \in \Sigma} P_w^{\sigma f}(\sigma_t | \boldsymbol{\sigma}_{0:t-1})}{P_e^f(\sigma_t | \boldsymbol{\sigma}_{0:t-1})} \Leftrightarrow$$

$$\Leftrightarrow \beta_{T'}^f = \prod_{i=1}^{k} \xi_i,$$

where $\xi_i > 1 \, \forall i = 1, .., k$. Therefore, $\beta_{T'}^f \to \infty$ as $k \to \infty$. Using Eqs. (8) and (1), $\alpha_{exp}^\emptyset := P_w^\emptyset(\boldsymbol{\sigma}_{0:T'}; M_{exp}) / P_w^\emptyset(\boldsymbol{\sigma}_{0:T'}; M) \to \infty$, and therefore there must be $k$ for which $\alpha_{exp}^\emptyset > \gamma$.

### A.2   Proof of Proposition 1

For each two nodes $p$ and $s$ such that $p$ is the parent of $s$, the change in the likelihood of the observed data according to $p$ induced by a change in the likelihood according to $s$ can be expressed as:

$$\alpha_{exp}^p = \frac{\bar{P}_w^p(\boldsymbol{\sigma}_{0:t})}{P_w^p(\boldsymbol{\sigma}_{0:t})} = \frac{1 + \alpha_{exp}^s \beta_t^p}{1 + \beta_t^p}. \tag{1}$$

Using Eq. (1), and applying the chain rule we have that:

$$\frac{d\alpha_{exp}^\emptyset}{d\alpha_{exp}^f} = \frac{d\alpha_{exp}^\emptyset}{d\alpha_{exp}^{p_1}} \frac{d\alpha_{exp}^{p_1}}{d\alpha_{exp}^{p_2}} \cdots \frac{d\alpha_{exp}^{p_{|f|-1}}}{d\alpha_{exp}^f}$$

$$= \prod_{d=0}^{|f|-1} \frac{\beta_t^{p_d}}{1 + \beta_t^{p_d}} =: \rho^f.$$

Since $\alpha_{exp}^\emptyset$ is linear in $\alpha_{exp}^f$, and considering an initial value of $\underline{\alpha}_{exp}^\emptyset = 1$, $\underline{\alpha}_{exp}^f = 1$ we have:

$$\frac{\alpha_{exp}^\emptyset - \underline{\alpha}_{exp}^\emptyset}{\alpha_{exp}^f - \underline{\alpha}_{exp}^f} = \frac{d\alpha_{exp}^\emptyset}{d\alpha_{exp}^f} \Leftrightarrow$$

$$\Leftrightarrow \alpha_{exp}^\emptyset = 1 + \rho^f \left( \alpha_{exp}^f - 1 \right).$$

## A.3 D2-CTW Pseudo-code and Auxiliary Subroutines

An online model-learning procedure for D2-CTW, which can be used directly either for symbolic prediction or reinforcement learning tasks, is sketched in Algorithm 1. Its most relevant operations are `getWeightedProb` (Algorithm 2), which outputs the probability of symbol $\sigma_t$, updates the predictive mixture accordingly, and evaluates the effect of potential fringe expansions and prune operations; and `postProcess` (Algorithm 3), which carries out those topological operations while respecting the overall memory bound, and updates the mixture as necessary to prevent biasing.

For model-based RL tasks, and during their planning stage, it may be necessary to sample repeatedly from the learned model without updating it (particularly for Monte-Carlo planning methods such as UCT). For this purpose of sampling without a corresponding update, Algorithm 2 can be trivially adapted by preventing symbol counts and $\beta$ from being updated for each node, and evaluating $P_w^\emptyset(\sigma|\cdot)$ for multiple $\sigma$.

---

**Algorithm 1** `train`

**Input:** Fringe depth $H$; likelihood test level $\gamma$; memory bound $L$
1: $\mathcal{T} \leftarrow initSuffixTree()$      // Contains model metadata *e.g.* size, root
2: $n^\emptyset \leftarrow getRoot(\mathcal{T})$
3: $\boldsymbol{\sigma} \leftarrow \emptyset$
4: **while** running conditions hold **do**
5:    Draw symbol $\sigma_t$ from source
6:    $d_{max} \leftarrow maxDepth(\mathcal{T}) + H$
7:    **if** $|\boldsymbol{\sigma}| > d_{max}$ **then**
8:      $\boldsymbol{\sigma} \leftarrow \boldsymbol{\sigma} \backslash \sigma_{t-d_{max}}$      // Keep only $d_{max}$ past symbols in context.
9:    **end if**
10:   $P_{D2-CTW} \leftarrow getWeightedProb(n^\emptyset, \mathcal{T}, \sigma_t, \boldsymbol{\sigma}, \gamma, H)$
11:   $\mathcal{T} \leftarrow postProcess(\mathcal{T}, L)$
12:   Use $P_{D2-CTW}$ *e.g.* for compression or control
13:   $\boldsymbol{\sigma} \leftarrow \boldsymbol{\sigma} \cup \sigma_t$
14: **end while**

---

**Algorithm 2** `getWeightedProb`$(n, \mathcal{T}, \sigma_t, \boldsymbol{\sigma}, \gamma, H)$

**Input:** Node pointer $n$; suffix tree $\mathcal{T}$; current symbol $\sigma_t$;
    context $\boldsymbol{\sigma}$; likelihood ratio threshold $\gamma$; fringe depth $H$
**Output:** Weighted conditional probability of $\sigma_t$
1: $P_e^n \leftarrow SADEstimator(n, \sigma_t)$      // Eq. (4)
2: $updateCounts(n, \sigma_t)$      // Update count vector
3: $d \leftarrow depth(n)$
4: $\sigma \leftarrow [\boldsymbol{\sigma}]_{|\boldsymbol{\sigma}|-d}$      // Get $d$−th to last symbol in context
5: **if** not $hasChild(n, \sigma)$ **then**
6:   **if** $fringeDepth(n) < H$ **then**
7:     $createFringeChild(n, \sigma, \mathcal{T})$
8:   **else**
9:     Return $P_e^n$
10:   **end if**
11: **end if**
12: $c \leftarrow getChild(n, \sigma)$
13: $P_w^c \leftarrow getWeightedProb(c, \mathcal{T}, \sigma_t, \boldsymbol{\sigma}, \gamma, H)$
14: $\beta_t^n \leftarrow updateBeta(\beta_{t-1}^n, P_e^n, P_w^c)$      // Eq. (7)
15: **if** $isFrontier(n)$ **then**
16:   $testExpansion(n, \gamma, \mathcal{T})$
17:   Return $P_e^n$
18: **else if** $fringeDepth(n) = 0$ **then**
19:   $testPrune(n, \mathcal{T})$
20: **end if**
21: $P_w^n \leftarrow P_e^n \frac{1+\beta_t^n}{1+\beta_{t-1}^n}$      // Eq. (6)
22: Return $P_w^n$

---

**Algorithm 3** `postProcess`$(\mathcal{T}, L)$

**Input:** Suffix tree $\mathcal{T}$; memory bound $L$
1: $f^*, L_{exp}, \alpha_{exp} \leftarrow getBestExpansion(\mathcal{T})$
2: $p^*, L_{prune}, \alpha_{prune} \leftarrow getBestPrune(\mathcal{T})$
3: **if** ($L < getSize(\mathcal{T}) + L_{exp} < L + L_{prune}$ **and**
     $p^*$ is not an ancestor of $f^*$ **and** $\alpha_{exp} \times \alpha_{prune} > 1$) **then**
4:    $pruneChildren(p^*, \mathcal{T})$
5: **end if**
6: **if** $L > getSize(\mathcal{T}) + L_{exp}$ **then**
7:    $expandFringe(f^*, \mathcal{T})$
8: **end if**

---

**Algorithm 4** `testExpansion`$(n, \gamma, \mathcal{T})$

1: **if** $\rho_t^n$ not known **then**
2:    $\rho_t^n \leftarrow updateRho(n, \beta_t^n)$            // Also updates values of $\rho$ for ancestors
3: **end if**
4: $\alpha_{exp}^{\emptyset} \leftarrow 1 + \rho_t^n \left( \frac{1+\beta_t^n}{2} - 1 \right)$
5: **if** $\alpha_{exp}^{\emptyset} > \gamma$ **then**
6:    $L_{exp} \leftarrow subtreeSize(n)$            // Size of the subtree below n
7:    $H_{exp}^n \leftarrow \frac{\log \alpha_{exp}^{\emptyset}}{L_{exp}}$
8:    **if** $H_{exp}^n > getBestExpansion(\mathcal{T})$ **then**
9:       $setBestExpansion(\mathcal{T}, n)$
10:    **end if**
11: **end if**

---

**Algorithm 5** `testPrune`$(n, \gamma, \mathcal{T}, L)$

1: **if** $\rho_t^n$ not known **then**
2:    $\rho_t^n \leftarrow updateRho(n, \beta_t^n)$            // Also updates values of $\rho$ for ancestors
3: **end if**
4: $\alpha_{prune}^{\emptyset} \leftarrow 1 + \rho_t^n \left( \frac{2}{1+\beta_t^n} - 1 \right)$
5: $L_{prune} \leftarrow subtreeSize(n)$            // Size of the subtree below n
6: $H_{prune}^n \leftarrow \frac{-\log \alpha_{prune}^{\emptyset}}{L_{prune}}$
7: **if** $H_{exp}^n < getBestPrune(\mathcal{T})$ **then**
8:    $setBestPrune(\mathcal{T}, n)$
9: **end if**

---

**Algorithm 6** `expandFringe`$(n, \mathcal{T})$

**Input:** Frontier node $n$
1: $\alpha \leftarrow \frac{1+\beta_t^n}{2}$
2: **while** $n \neq getRoot(\mathcal{T})$ **do**
3:    $n \leftarrow getParent(n)$
4:    $\alpha \leftarrow \frac{1+\alpha\beta_t^n}{1+\beta_t^n}$
5:    $\beta_t^n \leftarrow \alpha\beta_t^n$
6: **end while**
7: Clear frontier status at $n$

---

## A.4 Description of RL domains

We now provide further description of the environments that were used for our reinforcement learning experiments. Our implementation of the *T-Maze* and *partially observable mountain car* problems can be found at: `https://bitbucket.org/jmessias/po_gym`. For the *cheese maze* problem, we used the publicly available implementation of [4].

### A.4.1 T-Maze

In the *T-maze* environment (Fig. 1) [1], an agent starts at one end of a hallway (position $S$), where it observes the state of a *switch* (which can be up or down). The goal of the agent may be in one of the two possible positions at the other end of the hallway (positions $G$), with equal probability.

Figure 1: The *T-maze* environment.

The state of the switch identifies (deterministically) the position of the goal. The agent may take any of the four cardinal movement actions, which have deterministic outcomes. It can only uniquely observe the state of the switch, and its presence at the other end of the hallway – along the hallway, it receives a constant, uninformative observation, making this a non-Markovian problem. If the agent successfully reaches the position of the goal, it receives a reward of 100. Otherwise, if it enters the wrong end of the T intersection, it receives a penalty ($-5$). In both cases, the problem is reset and the goal position is re-sampled. The agent also receives a small penalty for each movement action ($-1$) and for bumping into walls ($-5$).

### A.4.2  Cheese Maze

The *cheese* or *M*-maze environment, introduced in [2] (and shown in Fig. 2, is a navigation problem where an agent must navigate to a goal position, but some of its observations are aliased. The environment has a total of elevn states, but only six distinct observations. The agent can perform the four cardinal movement actions deterministically. It receives an reward of 10 for reaching the goal, at which point the problem is reset; a reward of $-10$ whenever it bumps into a wall; and a reward of $-1$ otherwise.

Figure 2: The *cheese maze* environment.

### A.4.3  Partially Observable Mountain Car

Figure 3: The *mountain car* environment.

In the classic *mountain car* problem (depicted in Fig. 3) [3], a car must be driven up a steep hill, but does not have enough power to push itself up the goal from its initial position. Therefore, it must first pick up momentum by driving up a smaller hill opposite the goal. The agent may attempt to accelerate backwards, forwards, or stay neutral. For every time-step, it receives a small penalty ($-1$). The episode terminates when the agent reaches the goal, at which point the agent is reset.

In our partially observable version of this problem, the agent can observe its horizontal position, but not its velocity. We further discretized the position into 10 equally sized states. Otherwise, all problem-specific parameters are the same as in [3].

### A.4.4 Algorithm Parameters

In Table 1, we present the parameters that were used for our RL experiments across all algorithms and environments.

Table 1: Algorithm parameters for our RL experiments. $L$: maximum number of nodes in the model; $K$: maximum depth of the model; $\gamma$: D2-CTW fringe expansion threshold; $H$: D2-CTW fringe depth; $\epsilon_0$: initial exploration probability for $\epsilon$-greedy policy; $\epsilon_{dec}$: exponential decay rate on exploration probability; $S$: number of simulations in UCT; $C$: UCT exploration/exploitation tradeoff parameter; $h$: maximum horizon for UCT simulations. est: base estimator (Sparse Dirichlet or Frequentist).

| Environment | Algorithm | $L$ | $K$ | $\gamma$ | $H$ | $\epsilon_0$ | $\epsilon_{dec}$ | $S$ | $C$ | $h$ | est. |
|---|---|---|---|---|---|---|---|---|---|---|---|
| | D2-CTW | – | – | 1 | 2 | 0.2 | 0.9995 | 50 | 1 | 10 | SAD |
| T-Maze | CTW | – | 3 | – | – | 0.2 | 0.9995 | 50 | 1 | 10 | SAD |
| | PPM-C | – | 3 | – | – | 0.2 | 0.9995 | 50 | 1 | 10 | FREQ |
| | D2-CTW | 1000 | – | 10 | 2 | 0.2 | 0.999 | 100 | 2 | 10 | SAD |
| Cheese Maze | CTW | 1000 | 5 | – | – | 0.2 | 0.999 | 100 | 2 | 10 | SAD |
| | PPM-C | 1000 | 5 | – | – | 0.2 | 0.999 | 100 | 2 | 10 | FREQ |
| | D2-CTW | 1000 | – | 10 | 2 | 0.5 | 0.9995 | 50 | 1 | 10 | SAD |
| PO-Mountain Car | CTW | 1000 | 4 | – | – | 0.5 | 0.9995 | 50 | 1 | 10 | SAD |
| | PPM-C | 1000 | 4 | – | – | 0.5 | 0.9995 | 50 | 1 | 10 | FREQ |