[Reviews · NeurIPS 2017]

Reviewer 1



The paper develops a variation on Context Tree Weighting (CTW) which keeps memory costs low by adapting the depth of each branch to the extent that it aids prediction accuracy. The new algorithm, called Utile Context Tree Weighting (UCTW), is shown empirically in some illustrative examples to use less memory than fixed-depth CTW (since it can keep some branches short) and to be more effective under a memory bound (in which it must prune a node every time it expands a node). ---Quality--- As far as I can tell the technical claims and formalization of the algorithm are sensible. The experiments are, for the most part well designed to answer the questions being asked. One experiment that felt less well-posed was the T-Maze. The text says "We consider a maze of length 4. Thus we set K = 3." What does that "thus" mean? Is the implication that K = 3 should be deep enough to represent the environment? Later it says that "during the initial stages of learning the agent may need more than 3 steps to reach the goal." I assume that means that the agent might move up and down the "stem" of the T for a while, before reaching the goal, thus forgetting the initial observation if the suffix is limited to depth 3. If that's the case, then K = 3 is only sufficient to make predictions under the *optimal* policy, so it's no surprise that CTW+UCT can't perform well (UCT does random rollouts!). In fact, under those dynamics no finite suffix is enough to represent the environment (for arbitrary action sequences), so even the depth 4 model that UCTW learns is incorrect -- it just happens to be deep enough to be sufficiently robust to suboptimal behavior to allow the planner to work. I guess I'm just not entirely sure what to conclude from these results. We see that CTW does poorly when given inadequate depth (no surprise) and that UCTW adapts its depth, so that's fine. But UCTW doesn't learn a good model either, and it's basically a coincidence of the domain that it happens to work out for planning purposes. The other experiments, which are more focused on performance under memory bounds, make a lot more sense to me. ---Clarity--- I think the paper was pretty clearly written. The theoretical framework of CTW is always a challenge to present, and I think the authors have done pretty well. The main idea of the algorithm is described well at an intuitive level as well as at a formal level. I will say this: the name of the algorithm is confusing. The "utile" in utile suffix memory refers to the fact that the tree is expanded based on *utility* (i.e. value). The main point of that work was that the tree should only be as complicated as it needs to be in order to solve the control task. Here the tree is being split based on prediction error of the next observation, not utility, so it is strange to call it Utile CTW. I saw the footnote acknowledging and clarifying this mismatch...but the fact that you had to write that footnote is a pretty good sign that the name is confusing! How about "Incremental Expansion CTW", "Dynamic Depth CTW", or "Memory Bounded CTW"? UCTW is just not descriptive of what the algorithm does.... ---Originality--- Clearly the work is directly built upon existing results. However, I would say that it combines the ideas in a novel way. It re-purposes an alternative formulation of CTW in a clever way and develops the necessary updates to repair the tree after an expansion or pruning. ---Significance--- I think UCTW is interesting and may have a significant practical impact. CTW is an important algorithm in the compression literature and gaining interest in the AI/ML literature. I agree with the authors that memory is a major bottleneck when applying CTW to interesting problems, so a memory-bounded version is definitely of interest. Empirically UCTW shows promise -- though the experiments were performed on basic benchmarks they do demonstrate the UCTW uses less memory than fixed-depth CTW and can cope with a memory bound. UCTW is a little bit of a strange beast, though. One of the appeals of CTW is that it has this very clear Bayesian interpretation of representing a distribution over all prunings of a tree. It's not at all clear what happens to that interpretation under UCTW. UCTW is *explicitly* expanding and pruning the tree using a statistical test rather than the posterior beliefs. The claim that UCTW will eventually do as well as fixed-depth CTW makes sense, and is comforting -- it eventually finds its way to the original Bayesian formulation and can overcome any funkiness in the initialization due to the process up until that point. Furthermore it's not clear what happens to the regret bounds that CTW enjoys once this expansion/pruning scheme is introduced. This is not really an objection -- sometimes some philosophical/mathematical purity must be sacrificed for the sake of practicality. But it does make things muddier and it is harder to interpret the relationship of this algorithm to other CTW variants. Similarly, once we discard the clean interpretation of CTW, it does raise the question for me of why use CTW at all at this point? The authors raise the comparison to USM, but don't really compellingly answer the question "Why not just use USM?" The point is made that USM uses the K-L test, which is expensive, and doesn't have a memory bound. However, the main ideas used here (use likelihood ratio test instead and require a trade-off between expanding and pruning once the limit is hit) seem like they could just as easily be used in USM. I do no intend to suggest that the authors must invent a memory-bounded version of USM to compare to. However, if the authors do have a clear idea of why that's not a good idea, I think it would be valuable to discuss it. Otherwise I feel like the motivation of the work is a little bit incomplete. ***After Author Response*** I do think the name of the algorithm is misleading, and that leads to confusing comparisons too. For instance, in the author response the authors say "Also, the expansion tests in USM are performed over nonparametric representations of distributions over future reward, so the complexity of each test is a function of the sample size for each distribution." But, and I cannot stress this enough, *that is because USM is trying to predict value and UCTW is not.* They use different expansion tests because they address fundamentally different prediction problems. If one were to use a USM-like algorithm for predicting the next symbol from a finite alphabet, it would make perfect sense to represent the distribution using a histogram and use likelihood ratio tests instead of K-S; the complexity would be linear in the size of the alphabet, not the number of examples. USM uses K-S *because it is predicting a continuous value*. (In this case, I do nevertheless acknowledge that the CTW calculations have a nice side effect of making likelihood calculations efficient and thank the authors for that clarification). I think this paper should be accepted, so, if that happens, obviously it will be up to the authors what they do with the title and the algorithm name and so on. My point is that the direct link between USM and UCTW is not sound -- USM and UCTW are solving different problems. Pretty much the *only* thing UCTW takes from USM is the fact that it incrementally grows its depth. So it's fine to draw this connection between two algorithms that incrementally expand a suffix tree, and its good to acknowledge inspiration, but they can't be directly compared. At best you can compare UCTW to a USM-like algorithm that predicts future symbols rather than utility, but then, because it has a different prediction problem, the design choices of USM might not make sense anymore. I think the name UCTW reinforces this flawed direct comparison because at first glance it implies that UCTW is solving the same problem as USM, and it is not. None of this is fatal; a motivated reader can untangle all of this. I just hope the authors will get really clear about the distinctions between these algorithms and then make sure the paper is as clear as it can possibly be. I can see where the authors are coming from with the T-maze. I still think it's a bit of a wonky experiment, but adding a bit of the analysis given in the response to the paper would help a reader understand what the authors mean to extract from the results.